# Application of Graph Models to the Identification of Transcriptomic Oncometabolic Pathways in Human Hepatocellular Carcinoma

**DOI:** 10.3390/biom14060653

**Published:** 2024-06-03

**Authors:** Sergio Barace, Eva Santamaría, Stefany Infante, Sara Arcelus, Jesus De La Fuente, Enrique Goñi, Ibon Tamayo, Idoia Ochoa, Miguel Sogbe, Bruno Sangro, Mikel Hernaez, Matias A. Avila, Josepmaria Argemi

**Affiliations:** 1DNA and RNA Medicine Division, Applied Medical Research Center (CIMA), University of Navarre, 31008 Pamplona, Spain; sbarace@alumni.unav.es (S.B.); evasmaria@external.unav.es (E.S.); stefany.infante@udep.edu.pe (S.I.); sarcelus@unav.es (S.A.); 2Centro de Investigación Biomédica en Red de Enfermedades Hepáticas y Digestivas (CIBER-EHD), Av. Monforte de Lemos, 3-5. Pabellón 11, Planta 0, 28029 Madrid, Spainmaavila@unav.es (M.A.A.); 3Facultad de Medicina Humana, Universidad de Piura, Lima 15074, Peru; 4Bioinformatics Platform, Applied Medical Research Center (CIMA), University of Navarre, 31008 Pamplona, Spainmhernaez@unav.es (M.H.); 5Tecnun School of Engineering (TECNUN), University of Navarre, 31008 Pamplona, Spain; iochoal@unav.es; 6Liver Unit, Tecnun School of Engineering (TECNUN), University of Navarre, 31008 Pamplona, Spain; msogbe@unav.es; 7Instituto de Investigación Sanitaria de Navarra (IdisNA), 31008 Pamplona, Spain; 8Solid Tumor Program, Hepatology Laboratory, Applied Medical Research Center (CIMA), University of Navarre, C. de Irunlarrea, 3, 31008 Pamplona, Spain; 9Division of Gastroenterology Hepatology and Nutrition, University of Pittsburgh, Pittsburgh, PA 15232, USA

**Keywords:** hepatocellular carcinoma, RNA sequencing, metabolism, signature, graph, gene set enrichment analysis, gene set variation analysis

## Abstract

Whole-tissue transcriptomic analyses have been helpful to characterize molecular subtypes of hepatocellular carcinoma (HCC). Metabolic subtypes of human HCC have been defined, yet whether these different metabolic classes are clinically relevant or derive in actionable cancer vulnerabilities is still an unanswered question. Publicly available gene sets or gene signatures have been used to infer functional changes through gene set enrichment methods. However, metabolism-related gene signatures are poorly co-expressed when applied to a biological context. Here, we apply a simple method to infer highly consistent signatures using graph-based statistics. Using the Cancer Genome Atlas Liver Hepatocellular cohort (LIHC), we describe the main metabolic clusters and their relationship with commonly used molecular classes, and with the presence of *TP53* or *CTNNB1* driver mutations. We find similar results in our validation cohort, the LIRI-JP cohort. We describe how previously described metabolic subtypes could not have therapeutic relevance due to their overall downregulation when compared to non-tumoral liver, and identify N-glycan, mevalonate and sphingolipid biosynthetic pathways as the hallmark of the oncogenic shift of the use of acetyl-coenzyme A in HCC metabolism. Finally, using DepMap data, we demonstrate metabolic vulnerabilities in HCC cell lines.

## 1. Introduction

Gene set enrichment (GSE) methods have been widely used to facilitate the functional interpretation of transcriptomic data using sets of selected genes that are assigned to a specific biological context [1]. GSE methods such as gene set variation analysis (GSVA) have enabled the interpretation of thousands of gene expression changes between conditions or groups of patient samples by integrating statistical post hoc analysis into pathway-centric models [2]. However, the functional diversity between species, organs, tissues, and cell types as well as the heterogeneity of human cohorts weakens the generalization capabilities of most published signatures and gene sets, which were likely generated in highly controlled in vitro experiments on cell types and organs not related to the conditions under investigation. Similarly, some public gene sets have been curated by experts using knowledge about a specific pathway or biological process. For instance, the Metabolic Atlas (MetAtlas) repository was created from genome-scale metabolic models based on multi-omics and specific tissue subsystems [3]. For the Molecular Signature Database (MSigDB), hundreds of gene set collections were obtained largely from perturbation experiments [4], limiting their use in other, less controlled, transcriptomic analyses. A generalized method for adapting the available signatures to the biological context under study is thus warranted.

Hepatocellular carcinoma (HCC) is a leading cause of cancer-related death worldwide [5,6]. Hepatocyte differentiation is one of the most important prognostic factors in HCC [7], as exemplified by the histological classification first proposed by Edmondson and Steiner [8]. High-degree HCCs—Edmonson grades II to IV—behave aggressively, with an easily distinguishable atypical cell shape. As a highly proliferative cancer, the metabolism of high-histological-grade HCC shifts towards a more glycolytic phenotype, with more oxidative stress and glutathione usage and activation of the pentose phosphate pathway for the synthesis of purines and pyrimidines [9]. In recent years, the availability of transcriptomic data from human HCC has allowed the application of machine learning approaches to inferring metabolic classification with prognostic value [10,11,12,13]. These works have tried to understand the metabolic underpinnings of HCC in an unbiased manner, generating de novo signatures, mostly based only in cancer samples. Despite these analytic efforts, targeting cancer metabolic reprogramming is still an unmet objective. The performance of GSVA or other GSE methods using publicly available metabolic signatures has not been yet explored to define HCC metabolism, most probably because of the above-mentioned limitations.

In this work, to find metabolic vulnerabilities in human HCC, we developed a simple method to adapt published signatures by applying graph-based statistics to filter off-the-shelf gene sets before performing GSVA. We used the two largest available cohorts of sequenced HCC samples (TCGA-LIHC and ICGC-LIRI-JP) and showed the poor co-expression of published metabolic signatures present in MetAtlas and MSigDB.

The application of graph-based statistics led to the identification of metabolic clusters, with increased co-expression. We describe the association of newly generated metabolic signatures with other well-known transcriptomics HCC subclasses (such as those of Hoshida [14], Chiang [15]) and with the presence of *TP53* or *CTNNB1* driver mutations. We focus our study on signatures found to be enriched in tumors when compared to non-tumoral tissue, namely N-glycan, mevalonate and sphingolipid biosynthetic pathways. Finally, we show the genetic vulnerabilities within these pathways using the DepMap initiative (https://depmap.org/portal/, accessed on 15 April 2024) and suggest future avenues for targeting oncometabolic pathways in HCC.

## 2. Materials and Methods

### 2.1. Data Collection

Raw transcriptomic counts and the clinical information of the TCGA-LIHC cohort were obtained from Xenabrowser platform of the University of California Santa Cruz (UCSC) (https://xenabrowser.net/datapages/, accessed on 9 March 2023). RNA sequencing counts belonging to the ICGC-LIRI-JP cohort were downloaded from ICGC Portal (https://dcc.icgc.org/projects, accessed on 10 March 2023) with its corresponding clinical information. Clinical data from the TCGA-LIHC and the ICGC-LIRI-JP cohorts are summarized in Appendix A. In total, 559 HCC (359 from TCGA-LIHC and 200 from LIRI-JP) and 231 non-tumoral liver samples (49 from TCGA-LIHC and 172 from LIRI-JP) were included in the analyses. The patients’ cohorts were different regarding clinical characteristics. For example, LIRI patients were older, all Asian, and mostly chronically infected with HCV, while LIHC patients were half Caucasian–half Asian and mostly non-viral. Other differences included a higher G2 histological grade, a higher proportion of cirrhotic patients, and a higher proportion of CTNNB1 mutated tumors in LIRI, compared to LIHC. Human molecular signatures tested in this article were obtained from the Metabolic Atlas repository (https://metabolicatlas.org/explore/Human-GEM/gem-browser, accessed on 23 December 2021) and Molecular Signature Database (https://www.gsea-msigdb.org/gsea/msigdb/, accessed on 11 April 2024). Cell lines used for dependency and gene effect analyses were collected from the DepMap portal (https://depmap.org/portal/, accessed on 15 April 2024).

### 2.2. Normalization and Filtering Method

Raw counts from both cohorts were normalized using edgeR (version: 4.0.16) [16]. Genes with an expression lower than 1 count in more than 30% of samples were removed. Then, library size and normalization were performed via the calcNormFactors function and the TMM methodology. Counts per million (CPM) were calculated using the voom function of the limma package (version: 3.58.1) and used for downstream analyses.

### 2.3. Gene Set Adaptation Based on Graph-Based Statistics

Graph were developed using the igraph package (version: 2.0.2) [17]. Using CPM from normalized cohorts and retrieved metabolic gene sets, gene-gene co-expression was calculated using Spearman correlation after noticing extreme values in some observations. Correlation matrixes were filtered applying different correlation cutoffs called the Correlation Cutoff of Input Matrix (CCIM). For downstream analyses, a CCIM of 0.4 was used, as detailed in Section 3. Then, a graph was generated using graph_from_adjacency_matrix with mode = “undirected”. In these graphs, centrality was obtained using the eigen_centrality function, and Louvain communities were examined using the cluster_louvain function with resolution = 1. We considered eigenvalue centrality as our preferential metric for estimating the centrality of each gene in the graph in a range between 0 (isolated gene) and 1 (central gene). Since the purpose of our signature adaptation is to keep the biological meaning of the originally published gene set, only those communities with at least 20% of the original gene set size were included. In large gene sets, more than one community were found to have at least 20% of the original gene set, which led to the generation of more than one adapted gene set for some of the original signatures. To name these adapted signatures, therefore, the central gene of each community was used. Additionally, a core set of genes with the best centrality (eigenvalue ≥ 90% of the central gene eigenvalue) were defined.

To verify the persistence of the original meaning in the new adapted gene set, each novel signature was tested for its enrichment against the MSigDB and MetAtlas gene sets using the function enricher of the clusterProfiler package (version: 4.10.1) with the minGSSize set to 5 [18]. Any adapted gene that was not first-ranked in the hypergeometric test, meaning other signatures differing from the one that gave origin to the adapted one, was excluded from downstream analysis.

### 2.4. Sample Enrichment with ssGSEA

A single sample gene set enrichment algorithm (ssGSEA) from the corto package (version: 1.2.4) [19] was used to estimate the enrichment of each gene set across non-tumor and tumor samples. Those ones with less than 2 genes where removed from ssGSEA enrichment (min. size parameter set to 2).

### 2.5. Statistical Analysis Based on ssGSEA Score

Molecular and metabolic HCC subgroups obtained from previous works [10,14,15] and sample information presented in clinical data were evaluated considering the ssGSEA score. Means and standard deviations were found and statistical tests were performed and retrieved for further visualization.

### 2.6. Ridgeplot and Heatmap Visualization

Statistically significant gene sets found in molecular and clinical categories were plotted using pheatmap (version: 1.0.12) [20], ggplot2 (version: 3.5.0), ggridges (version: 0.5.6) and ggthemes (version: 5.1.0) [21]. When analyzing differences in ssGSEA scores between non-tumor and tumor samples, only paired samples were selected.

### 2.7. Survival Analysis

TCGA patients were previously divided into two partitions (training and validation subsets) with a proportion of 50:50. ICGC-LIRI-JP was considered as test subset in survival analyses. Survival analysis was performed by taking the median ssGSEA score as a numerical discriminator between high- and low-expression groups. The median survival of both groups was estimated using the survival (version: 3.5-5) and survminer (version: 0.4.9) packages [22,23]. In addition, median survival difference (MSD) was computed by subtracting low- from high-expression samples.

### 2.8. Estimation of Gene Dependency and Gene Effect with DepMap Portal

Cell lines from HCC (Appendix A) were analyzed in silico to estimate the effect and dependency of each cell line on the knock-out of the genes included in the relevant adapted signatures. To this end, we downloaded the CRIPR-Cas9 screen from the DepMap portal [24,25]. Briefly, a gene dependency score of 0 means minimal dependency, whereas one indicates full dependency of the cell line on the specific gene. Regarding gene effect score, a score of 0 represents no effect of the deletion of that particular gene on cell survival; negative scores indicate a deleterious effect; and positive scores indicate a protecting role of that particular gene on the cell line.

### 2.9. Statistical Analysis

A Shapiro–Wilk test was used to test the normality of each distribution. For variables with two groups, a *t*-test or Wilcoxon test was performed according to normality tests. For three or more groups, parametric ANOVA or Kruskal–Wallis tests were performed, and each individual comparison were also evaluated two by two. A log-rank test was used for survival analyses. In box plots, median and interquartile range were displayed. Only *p*-values less than 0.05 were considered statistically significant.

## 3. Results

### 3.1. Graphs Generate Highly Compacted Metabolic Signatures

Considering metabolism as a crucial hallmark of HCC development, we first investigated how co-expressed curated signatures from the Hallmark collection of MSigDB are in the context of the HCC transcriptome (Figure 1A). As expected, genes belonging to proliferative signatures such as E2F_TARGETS, describing a high protein translation rate, or MYC_TARGETS—as 60% of HCCs in TCGA overexpress the MYC oncogene—have high median gene-to-gene correlation (MGGC). Metabolic signatures present moderate to low co-expression, with MGGC values closer to non-hepatic non-cancer signatures, which present the lowest co-expression. When the Metabolic Atlas (MetAtlas) repository was tested, similarly low levels of MGGC were found (Figure 1B), with higher values in signatures related to oxidative phosphorylation or beta-oxidation of fatty acids when compared to others such as acylglyceride metabolism or cytosolic carnitine shuttle. These data confirm the specificity of the biological context of HCC and suggest that enrichment scores based on these signatures could be affected by a low signal-to-noise ratio.

We thus designed a pipeline to derive highly co-expressed signatures from public gene sets using graph-based statistics, aiming to identify networks of genes based on co-expression matrices from human HCC transcriptomic data (see Section 2: Methods). To pursue this, we retrieved 130 metabolic signatures from Metabolic Atlas and 315 Hallmark (H) and canonical pathways from curated gene sets (C2-CP) from MSigDB (Figure 1C). A total of 445 metabolic signatures were analyzed with 1684 genes shared in both databases, 2105 uniquely present in MetAtlas, and 717 specific to MSigDB. As expected, globally, adapted gene sets obtained from bioinformatic pipeline presented better MGGC compared to their original counterparts (Figure 1D).

Next, TCGA-LIHC samples were randomly clustered into training and validation sets, while ICGC-LIRI-JP composed the test set. Graphs generated with higher correlation thresholds generated less populated graphs and smaller Louvain communities (Figure 1E,F). Testing all possible correlation thresholds, we detected that a range between 0.2 and 0.5 generated the highest number of communities with the highest number of co-expressed genes, with a considerable increase in MGGC and decrease in median gene–gene variance (MGGV) at the expense of a limited reduction in gene set size to between 40 and 30% of its original size, indicating that a significant number of genes in the original public signature in MetAtlas or MSigDB were preserved (Appendix A). We also confirmed that all newly generated signatures were ranked first after a hypergeometric test was performed against the universe of all available gene signatures (all GSEA H and C2 collections plus MetAtlas signatures). The final collection of gene signatures was named after its central gene (with an eigenvalue of 1 or the highest eigenvalue in the case of one original signature generating two or more new Louvain communities, see Methods). Finally, among those signatures with the same central gene, some of them from different signatures and representing similar metabolic pathways were merged.

### 3.2. Metabolic Clusters Are Tumor-Specific and Associated with Molecular Subtypes in the TCGA LIHC Cohort

The intermediate metabolism is one of the most important functions of normal hepatocytes. We hypothesized that tumor communities could not be entirely coincident with non-tumor ones, regarding both the number and size of the gene sets and the specific central genes. It was found that non-tumor communities presented higher number of clusters, unique genes per cluster, unique core genes, and unique central genes when compared to tumor ones (Figure 2A,B) and that the central genes defining clusters were mostly divergent (Figure 2C–F). After applying gene set adaptation and selecting enriched gene sets, 74 different signatures were obtained, conforming to different metabolic signatures, of which only 17 were also found in non-tumor samples (Figure 2F, Appendix A).

We then decided to explore whether the relative increase or decrease in the global expression of a specific community defined a particular biology or was associated with a previously described HCC subtype. We thus interrogated the enrichment of the new graph-identified signatures using ssGSEA in the HCC samples of the TCGA-LIHC cohort (Figure 2G–J). The hierarchical clustering of the signatures led to four main groups of pathways (Figure 2G, Table 1), which are described as follows.
The largest group of pathways included a varied array of typically hepatic metabolic functions, some related to fatty acid metabolism and transport, such as Cytochrome P450 Family 4 Subfamily A Member 22 (CYP4A22) or Carnitine Palmitoyltransferase 2 (CPT2), which is involved in mitochondrial long-chain fatty acid transport. Others related to the catabolism of amino acids, such as Glutaryl-CoA Dehydrogenase (GCDH), an important enzyme in the degradation of lysine, hydroxylysine, and tryptophan; Sarcosine Dehydrogenase (SARDH) involved in glycine cleavage; Alpha-Aminoadipic Semialdehyde Synthase (AASS), in charge of lysine degradation; and Methylcrotonoyl-CoA Carboxylase 2 (MCCC2), involved in the catabolism of leucine. Some additional pathways related to this group included those centered in enzymes of the respiratory chain, such as the subunits of the Succinate Dehydrogenase (SDHA and B) and enzymes and transporters involved in the processing of drugs and xenobiotics (CYP3A4, AOX1, NAT2, and ABCC2).The second largest group of pathways included functions related to metabolic aspects of extracellular matrix (ECM) organization and cell adhesion, such as signatures centered in Lumican (LUM), Decorin (DCN), Versican (VCAN), thrombospondin 2 (THBS2) and Collagen Type III Alpha 1 Chain (COL3A1); pathways related to inflammation, including the leukotriene biosynthesis pathway centered in Arachidonate 5-Lipoxygenase Activating Protein (ALOXAP5); and pathways related to the modification of glucosamine glycans, such as those centered in the Carbohydrate Sulfotransferase 3 (CHST3) and the Beta-1,3-Galactosyltransferase 5 (B3GALT5) genes.A third group was composed of signatures related to nucleotide synthesis, such as the Nucleoside Diphosphate Kinase 1 (NME1), protein synthesis (RPL17A-driven signature), mitochondrial function (COX5B), glutathione (GPX4) and cytoplasmic glycosylation pathways (GMPPA).The fourth group included gene sets related to mevalonate and cholesterol biosynthesis, such as Isopentenyl-diphosphate Delta Isomerase 1 (IDI1), Farnesyl Diphosphate Synthase (FDPS), 7-Dehydrocholesterol Reductase (DHCR7), and the Emopamil-Binding Protein (EBP).Interestingly, a residual group with ssGSEA values unrelated to any of the four mentioned groups encompassed transcriptional regulators and nuclear factors such as the ones included in the mediator complex and nuclear co-repressors included in the EP300 community, and a PIK3C2A-centered signature, with genes involved in inositol-phosphate metabolism.

Interestingly, some of the Hoshida subtypes originally not defined based on metabolic characteristics [14] clustered according to some of the new graph-based metabolic signatures (Figure 2G, top). For example, while Hoshida S1 class was enriched in group 2 signatures (metabolism of ECM-related proteins), S3 class had higher scores of group 1 (liver-specific). Both signatures are present in the non-tumoral tissue, which suggests that in S1 tumors, there is a specific downregulation of group 1 signatures, such as CYP3A4 and SDHA (Figure 2H), and in S3 tumors, there is a downregulation of group 2 signatures, such as VCAN or B3GALT5 (Figure 2I). As expected, the intensity of liver-specific signatures was higher in non-tumoral livers than in S3 tumors (Figure 2H). On the other hand, the intensity of ECM signatures was slightly higher in S1 tumors than in non-tumors (Figure 2I). Hoshida class S2 had lower expression of both group 1 and group 2 signatures when compared to non-tumor samples (Figure 2G–I). Whether group 1 and group 2 signatures reflect two differentiation states is unknown. Two groups of metabolic signatures (group 3, “proliferation” and group 4, “cholesterol”) had increased ssGSEA scores in tumor samples when compared to non-tumoral livers, indicating potential metabolic targets. Samples with high scores in group 3 and 4 signatures were distributed among Hoshida S1, S2 and S3 classes. This could indicate that while liver differentiation and ECM define one layer of metabolic subtype, the proliferation rate—in terms of protein synthesis and nucleotide and mitochondrial metabolism—and the induction of the mevalonate pathway could define a second perhaps more dynamic onco-metabolic shape (Figure 2J). Interestingly, not all tumors with high group 3 scores had increased group 4 signature scores, indicating different mechanisms of regulatory control.

We then analyzed Chiang transcriptomic classes [15] in the context of the newly generated metabolic signatures. As with Hoshida subclasses, there were metabolic differences between Chiang subclasses. For example, the proliferative subgroup presented the lowest levels of expression of group 1 signatures, whereas CTNNB1, Polysomy 7 and Interferon classes presented higher expression of these signatures (Appendix A). Conversely, when interrogating group 2 signatures, CTNNB1 and Polysomy 7 seven (but not the Interferon class) had the lowest level of expression (Appendix A). These results indicated the relationship between a specific molecular subtype and the metabolic status of the cell.

Finally, the metabolic landscape of HCC, as inferred by the generated signatures, matched with previously described metabolic classes. Bidkhori metabolic class iHCC1 [10] was enriched with liver-specific (group 1) signatures (Appendix A), while iHCC3 tumors had the lowest levels of these signatures but the highest levels of metabolism of ECM (Appendix A). As described, the iHCC2 metabolic subgroup had an intermediate score in group 1 signatures when compared to iHCC1 and 3. Some signatures, such as those related to drug and xenobiotic metabolism (NAT2, UGT1A4) and those related to steroid metabolism (HSD17B4) (Appendix A), were enriched specifically in iHCC2 samples.

### 3.3. TP53 and CTNNB1 Mutant Tumors Are Metabolically Diverse

HCC’s main driver mutations include deleterious *TP53* variants, activating N terminal *CTNNB1* mutations and activating variants of the *TERT* gene promoter. Among them, *TP53* and *CTNNB1* mutations are mutually exclusive in most HCC patients, which allows for the comparative analysis of their specific biologic behavior. With newly adapted signatures, we could define a specific metabolic shape for *TP53-* and *CTNNB1*-mutated tumors. Tumors bearing *TP53*-null mutations presented overexpression of the NME1 signature related to purine metabolism. On the other hand, *CTNNB1*-mutated patients presented an enrichment in metabolic signatures such as *MAT1A* and *CYP3A4*, when compared to *TP53*-null patients. Conversely, tumors with wild-type *TP53* presented higher metabolic enrichment of *MAT1A* and *CYP3A4*. These data suggest that *CTNNB1* mutation supports the maintenance of a liver metabolic-like phenotype in HCC, while *TP53*-mutant tumors are more de-differentiated and highly proliferative (Appendix A).

### 3.4. The Survival of Patients with Low Metabolic Tumors Is Worse in the LIHC and LIRI Cohorts

It has previously been observed that Bidkhori iHCC1 class determines survival prognosis [10]. We therefore used the training, validation, and test cohorts to verify the prognostic significance of our derived metabolic signatures (Figure 3A). Only liver-specific metabolic signatures (group 1) such as ABAT, DMGDH and GLYAT, which were downregulated in tumors (Figure 3B), were associated with prognosis in all three cohorts. Patients with tumors with higher expression of these metabolic signatures had increased overall survival (Figure 3C). This result indicates the validity of the signatures found in the LIHC cohort in interrogating the metabolic phenotype of unseen data, such as the LIRI-JP cohort. The median survival difference was lower in the LIRI-JP cohort, perhaps due to the better global survival in this cohort.

### 3.5. Mevalonate, N-Glycan and Sphingolipid Biosynthesis Pathways Shape Tumor Metabolism in Human HCC

We then decided to study in more detail the few signatures with increased scores in tumors when compared to non-tumoral livers in both cohorts LIHC and LIRI, regardless their molecular subtype (signature groups 3 and 4). Some signatures, such as IDI1 signature, glycosylation (GMPPA signature), sphingolipid (SPTLC1 signature), and nucleotide (NME1 signature) metabolism, and the catabolism of the polyamines were related to the mevalonate/cholesterol biosynthetic pathway with mostly proteasome subunit genes (PSMB3 signature). Isopentenyl-diphosphate delta-isomerase 1 (*IDI1*) was the most centric gene of the Louvain community with increased enrichment scores (Figure 4A) involving other cholesterol-related genes such as Squalene Epoxidase (*SQLE*), Mevalonate Diphosphate Decarboxylase (*MVD*), Sterol Regulatory Element-Binding Transcription Factor 2 (*SREBP2*), Farnesyl Diphosphate Synthase (*FDPS*), and Phosphomevalonate Kinase (*PMVK*), all of them with increased expression in the HCC of both the LIHC and the LIRI-JP cohorts (Figure 4B,C).

GDP-mannose pyrophosphorylase A, encoded by GMPPA gene, was the central gene in a signature associated with other enzymes related to the synthesis and ramification of N-Glycans in the cytoplasmic and luminal domains of the endoplasmic reticulum wall, including Dolichyl-Phosphate Mannose Synthase Subunit 3 and 8 (ALG3, ALG8), Dolichol kinase (DOLK), and Required for FTase Activity Protein 1 (RTF1), the flippase that internalizes the glycans to be incorporated to nascent polypeptides inside the ER. All of these genes were upregulated in tumors both in LIHC and LIRI (Figure 4D,F).

The signature centered in the Serine Palmitoyltransferase Long-Chain Base Subunit 1 (*SPTLC1*) was also upregulated in tumors of both the TCGA-LIHC and ICGC-LIRI-JP cohorts (Figure 4G). SPTLC1 is a key enzyme in sphingolipid biosynthesis, catalyzing the generation of ketosphingoids from Serine and Acetyl-CoA, the rate-limiting step for the generation of ceramides. Other enzymes in the same signature belonging to the lysosomal pathway of ganglioside catabolism, such as Hexosaminidase Subunit Beta (*HEXB*) and Neuraminidase 1 (*NEU1*), were also overexpressed in tumor samples of both LIHC and LIRI-JP (Figure 4H–I).

The NME1 signature comprised nucleotide metabolism and polimerase enzymes related to DNA replication and transcription. As expected, tumor samples presented higher expression in comparison with non-tumor ones (Appendix A). Among the most relevant genes in this signature were Thymidine Kinase 1 (*TK1*), Uridine-Cytidine Kinase 2 (*UCK2*), and Deoxythymidylate Kinase (*DTYMK*), all of which were involved in nucleotide metabolism (Appendix A).

Finally, overexpression of the PSMB3 signature, was related to the metabolism of polyamines (but also to the more general proteasome function) and was found to differentially increase in tumor samples (Appendix A). Several components and subunits of the proteasome presented a general upregulation in tumor samples of both the LIHC and LIRI-JP cohorts (Appendix A).

### 3.6. HCC Metabolic Vulnerabilities in Mevalonate, N-Glycan, and Sphingolipid Pathays as New Targets for Therapy

The aforementioned results point to a possible implication of mevalonate/cholesterol, N-glycan, and sphingolipid metabolism in HCC biology, regardless of the molecular subtype or the driver mutation. Since these pathways are induced in tumor samples when compared to non-tumoral liver, we thought they could constitute potential targets for anticancer therapy. Thus, to determine the importance of these enzymes for the survival of HCC cells, DepMap data were used to analyze cell viability and dependency when these genes are targeted. After analyzing six tumor-specific signatures (IDI1, GMPPA, NME1, PSMB3, SPTLC1 and EBP), the gene effect and gene dependency were measured in different non-cancerous cell lines and liver cancer lines of human HCC. The genes 3-Hydroxy-3-Methylglutaryl-CoA Synthase 1 (HMGCS1), 3-Hydroxy-3-Methylglutaryl-CoA Reductase (HMGCR), Farnesyl Diphosphate Synthase (FDPS), and Mevalonate Diphosphate Decarboxylase (MVD) were the most affected genes upon CRISPR-knock out, impacting cell survival as exemplified by the highest gene dependency scores (Figure 5A) and the most negative gene effect (Figure 5B). Interestingly, the knock-down of another gene in the same community, the NAD(P)H Steroid Dehydrogenase-Like (NSDHL), conferred higher survival and had a positive effect on cancer cells (Figure 5B).

In the N-Glycan GMPPA signature, the UDP-N-Acetylglucosaminyltransferase Subunit (ALG14) and RFT1 Gene (RFT1) accounted for the highest vulnerability in most HCC cell lines tested (Figure 5C,D). In the case of SPTLC1 signature, some but not all HCC cell lines were dependent on SPTLC1 and SLC33A1 genes, indicating a pathway less relevant for HCC survival than the previous two.

## 4. Discussion

In the present work, we implement a simple computational method for inferring metabolic pathways, using public signatures in MSigDB and the MetAtlas, by understanding HCC-specific gene networks using graph-based statistics. The newly generated gene communities are highly co-expressed and represent major metabolic domains of liver cancer cells. We use ssGSEA as an enrichment method to infer the activity of these pathways in samples of the two largest cohorts with available transcriptomic data, the TCGA-LIHC and the ICGC-LIRI-JP, including a total of 559 HCC samples and 149 non-tumoral liver samples.

We show that the metabolic phenotype is only partly associated with previously described signatures such as Hoshida and Chiang. As expected, liver-enriched metabolic pathways were associated with Hoshida S3 subclass and Chiang subclasses CTNNB1, Poly7, and Interferon. On the other hand, ECM metabolism was associated with Hoshida S1. We nevertheless describe the presence of two groups of signatures, groups 3 “proliferation” and 4 “Mevalonate/cholesterol”, which can be increased or decreased in Hoshida subclasses S1, S2 and S3, indicating a level of metabolic regulation that works in asynchrony with regard to hepatocyte differentiation. In conclusion, our work confirms and enriches previous transcriptomic classification of HCC, adding an important validation of the main findings in the LIRI-JP cohort, which so far has not been used to study metabolic profiles. We show that *TP53*-null and *CTNBB1*-mutated tumors have divergent metabolic profiles, which is consistent with what has been previously described [26,27,28].

The strength of this study is our focus on signatures enriched in tumoral samples when compared to non-tumor tissues. Previous works [10,14,15] have depicted the transcriptomic landscape of HCC but did not consider the expression of metabolic signatures in non-tumoral livers. Here, we show that those tumors with high liver-specific signature enrichment are still poorly differentiated when compared to non-tumoral livers. This approach helps us discover metabolic pathways increased in tumors that constitute part of the hallmarks of liver cancer and that could be targeted by future synergistic approaches using immunotherapy.

We confirm that metabolic pathways related to nucleotide biosynthesis, such as the NME1 signature, are related to highly proliferative tumors and have a role in HCC progression [29], although this role could be non-HCC-specific but common to other cancer types. We describe the mevalonate, the N-glycan, and sphingolipid biosynthetic pathways as induced pathways in HCC and thus potential targets for therapy. Although these new findings validate previous evidence of the role of these pathways in cancer cell survival and immune evasion, such data are so far lacking and fragmented in the literature on HCC.

Regarding mevalonate biosynthesis, it has been shown that IDI1 promotes tumor growth [30]. Interestingly, IDI1 represses CCL5- and CXCL10-expressing cells in the tumor microenvironment, increasing the capacity for immune evasion. On the other hand, EBP inhibitors have been shown to impair prostate cancer proliferation [31]. FDPS has been largely studied in other cancers. For instance, its role in promoting glioblastoma growth is known; it acts by recruiting tumor-associated macrophages through increased expression of CCL20 [32]. This same immunosuppressive mechanism set up by cancer cells has been demonstrated in in mouse models of beta-catenin-induced HCC [33]. In osteosarcoma cell lines and HeLa cells, FDPS was also able to change the ECM organization and promote proliferation and DNA repair [34]. Finally, FDPS has been proposed as a biomarker of breast cancer development [35]. Squalene epoxidase (SQLE) is capable of promoting tumor growth by inhibiting apoptosis [36] and is able to interact with the TGFb-SMAD axis to promote EMT and metastatic capacity [37]. We observed using the DepMap data that HMGCR, HMGCS1, FDPS, MVD and IDI1 confer different degrees of vulnerability when knocked out with CRISPR-Cas9 in several human HCC cell lines. Whether available HMGCR inhibitors such as statins could be used as repurposed drugs for combination with immunotherapy remains a provocative possibility.

Few glycosylation pathways have been described in HCC. An abnormal glycosylation of the ectonucleotidase CD73 was found in HCC samples [38] but not in adjacent livers. More broadly, glycosylation patterns are known to be present in a variety of cancer types and contribute to their fitness and evasion from the immune surveillance [39,40,41]. Targeting ALG14 or RFT1 led to HCC cell death in a consistent fashion, expanding a variety of cancer cell lines. To determine whether targeting these genes or other members of this pathway is feasible and non-toxic for non-tumoral cells, further preclinical work is required.

Finally, for the sphingolipid biosynthesis pathway, conflicting data have been reported. On one hand, the SPTLC1 gene has been found to be anti-oncogenic. In colorectal cancer, the low expression of SPTLC1 leads to worse prognosis [42]; in renal cell carcinoma, it inhibits cell proliferation [43], and in lymphoma patients, a mutation of SPTLC1 increases enzymatic activity, thereby sensitizing BCR-ABL tumors to imatinib [44]. On the other hand, serum ceramides and sphingolipids such as S1P and SA1P are increased in patients with HCC but not in cirrhotic controls [45], and the blockade of sphingolipids in Huh7 and HepG2 cell lines leads to increased susceptibility to sorafenib [46]. More broadly, it has been described that sphingolipids are produced in higher amounts in cancer cells and that the sphingosine-1-phosphate (S1P) intermediate promotes proliferation, migration, and EMT [47] and regulates the interphase with other cells through the inhibition of sindecan 1 [48].

One potential unifying model to explain the relationship between the above-mentioned metabolic pathways in HCC could be the cancer-specific change in the use of Acetyl-CoA, the most-used substrate of the cell for anabolic and energetic functions (Figure 6). In non-cancerous liver cells, Acetyl-CoA is a central metabolic intermediate, and the maintenance of the Acetyl CoA pool is essential for growth, proliferation, and protein modification. Cancer cells have developed the capacity to capture acetate as an alternative source to glucose from the circulation and even from the intestinal microbiome [49,50]. In the present work, we show that in human HCC, many of the metabolic pathways using Acetyl-CoA—such as lipid biosynthesis—are downregulated when compared with not-tumoral tissues. This is particularly evident in highly de-differentiated tumors, while unique genuinely overexpressed metabolic signatures are the mevalonate/cholesterol, N-glycan, and sphingolipid pathways, all three meant to deviate Acetyl-CoA precursors into pro-tumoral biosynthetic pathways related to protein glycosylation, turnover, and ECM organization. The fact that the main transporter of Acetyl-CoA into the lumen of ER and lysosomes, SLC33A1, is one of the most overexpressed genes in HCC may lead to its experimental evaluation as a potential target for HCC therapy.

Patients with advanced HCC are currently treated with immunotherapy as a first line, and two combinations are currently approved in Western countries and in Asia for this indication: atezolizumab plus bevacizumab and tremelimumab plus durvalumab [51,52]. One of the main limitations of this work is the lack of baseline liver and tumor transcriptomic data from patients with advanced HCC treated with these regimens. Both the TCGA-LIHC and ICGC-LIRI-JP cohorts mostly include patients with early HCC that were treated through curative therapies such as resection. We have not found a clear association of immune infiltration with metabolic signatures. Whether the inhibition of the metabolic pathways found in the present work could impact the response of these patients to immunotherapy is an area for further investigation. Additionally, the role of these signatures as prognostic or predictive biomarkers is yet to be explored.

## 5. Conclusions

Utilizing computational methods to infer gene networks specific to HCC, this study identifies major metabolic domains in liver cancer cells, validating and enriching previous transcriptomic classifications. The metabolic phenotype of HCC reveals associations with known HCC subclasses, such as Hoshida and Chiang, while also highlighting the presence of distinct metabolic pathways in poorly differentiated tumors compared to non-tumoral liver tissue. Mevalonate/cholesterol, N-glycan, and sphingolipid pathways emerge as potential therapeutic targets for HCC, with specific genes within these pathways showing promise for targeted therapies, potentially in combination with immunotherapy. This study also provides insights into the functional roles of various metabolic pathways in HCC progression, including nucleotide biosynthesis, glycosylation, and sphingolipid biosynthesis, shedding light on their reported involvement in tumor growth, immune evasion, and metastasis. The findings raise questions about the impact of targeting these metabolic pathways when treating advanced HCC patients and suggest further investigation into the potential use of these metabolic signatures as prognostic or predictive biomarkers. Additionally, this study underscores the need for transcriptomic data from patients undergoing immunotherapy for a more comprehensive understanding of treatment responses when using the current standard of care.

## Figures and Tables

**Figure 1 biomolecules-14-00653-f001:**
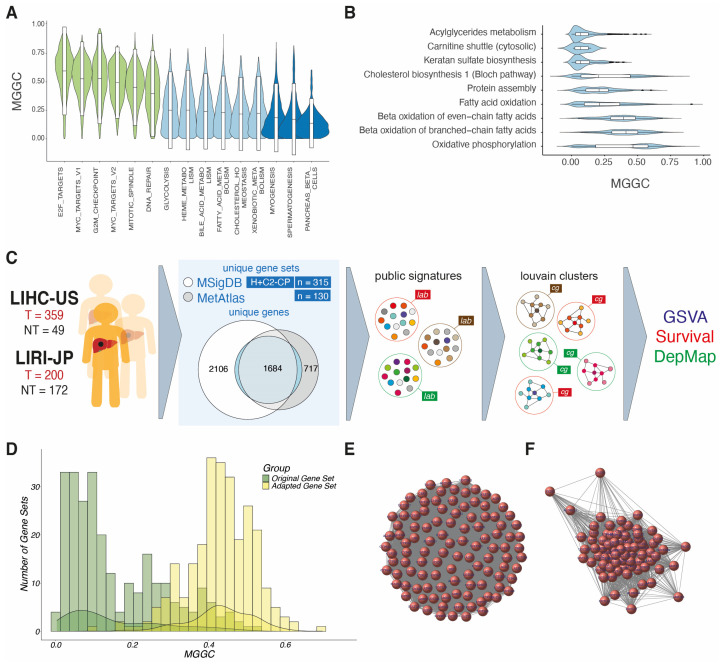
Using graphs to adapt public signatures. (**A**) Violin plot showing the median gene to gene correlation (MGGC) of selected Hallmark signatures in the LIHC HCC cohort, including those related to proliferation (green), metabolism (light blue), and unrelated to liver or liver cancer (dark blue). (**B**) MGGC of signatures from the Metabolic Atlas (MetAtlas). (**C**) Scheme of the method of adaptation of public signatures from the Molecular Signature Database (MSigDB) and MetAtlas to identify centric nodes and metabolic clusters using graphs. (**D**) Effect of the method on the MGGC of metabolic signatures from MSigDB and MetAtlas in LIHC. (**E**) An example of a non-filtered co-expression matrix of “Xenobiotic Metabolism” signature of the MetAtlas in the LIHC cohort, where all genes (nodes) are connected in an apparently equal relationship (edge). (**F**) An example of a Louvain cluster obtained by after graph-based adaptation was applied to the “Xenobiotic Metabolism” signature.

**Figure 2 biomolecules-14-00653-f002:**
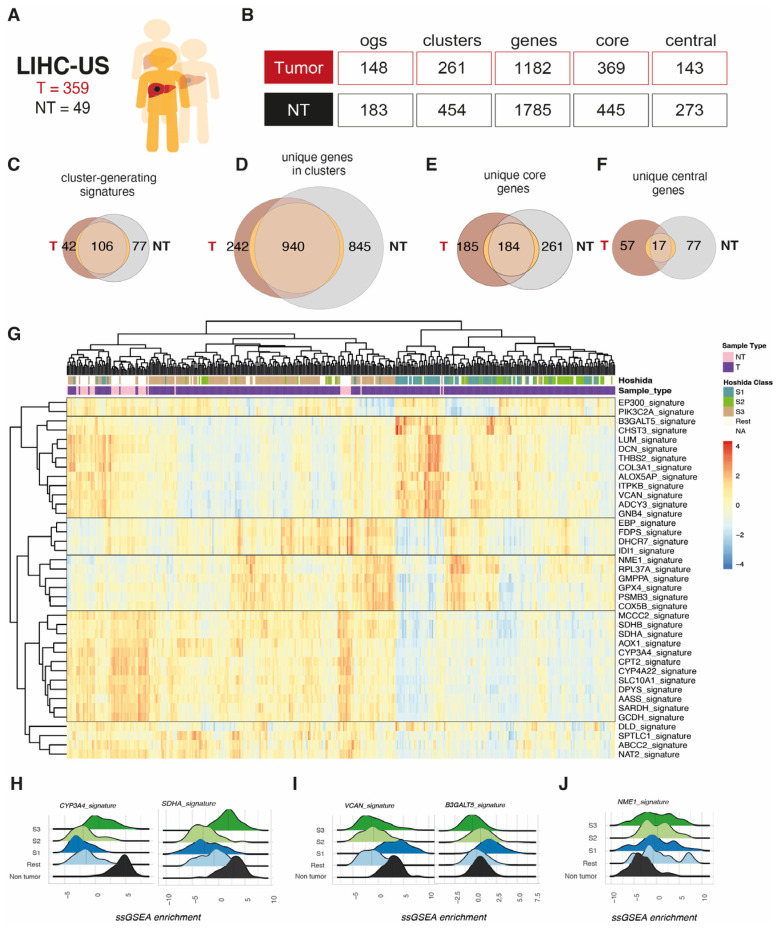
Identification of metabolic clusters in HCC and their association with transcriptomic classes. (**A**) LIHC was used as the training cohort, where tumor (HCC, *n* = 359) and non-tumor (NT, *n* = 49) samples were analyzed. (**B**) From the original MSigDB and MetAtlas signatures, unrestricted co-expression matrix (r threshold 0.05) led to the identification of only 148 metabolic clusters in HCC and 183 in NT, which increased to 261 and 454 in HCC and NT, respectively, with an r threshold of 0.4, as used in downstream analyses. These clusters included 1182 and 1785 unique genes, 369 and 445 core genes, and 143 and 273 central genes in HCC and NT, respectively (see Section 2). (**C**–**F**) Overlap between signatures (**C**), unique genes (**D**), unique core genes (**E**), and unique central genes (**F**) found in HCC and NT. (**G**) Heatmap of ssGSEA scores using newly identified metabolic clusters and their association with Hoshida classes S1, S2, and S3. (**H**–**J**) Ridge plots showing the expression of signatures belonging to group 1, 2, and 3 by Hoshida class S1, S2 and S3.

**Figure 3 biomolecules-14-00653-f003:**
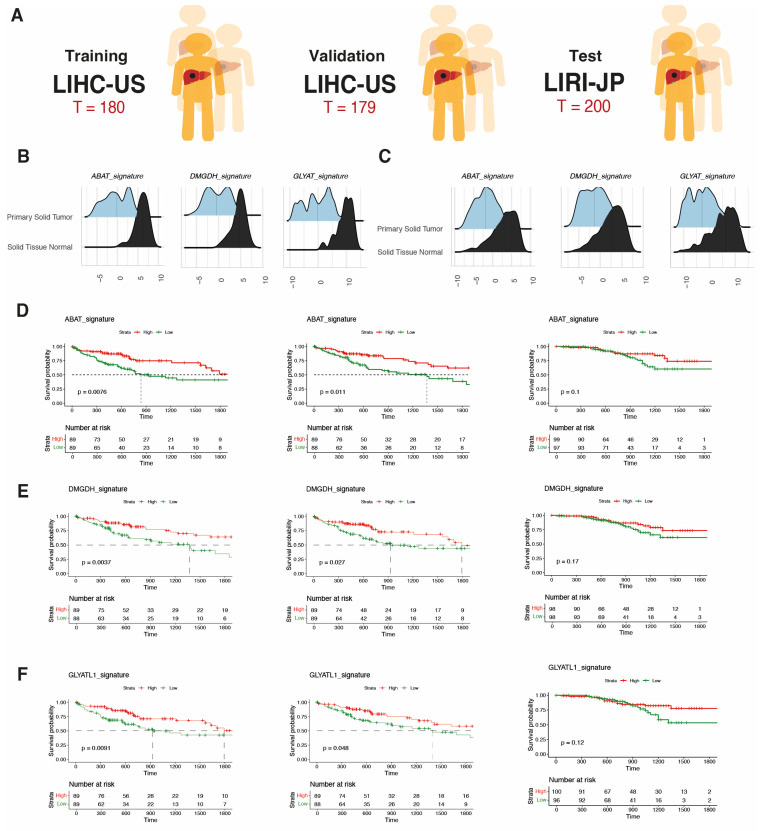
Validation of the method for prognostic prediction in patients with HCC. (**A**) A random 50:50 split of the LIHC cohort led to the training (*n* = 180) and validation (*n* = 179) cohorts for prognostic analyses, while the LIRI-JP cohort (*n* = 200) was used as the test cohort. (**B**,**C**) Overall ssGSEA scores of prognostic signatures ABAT, DMGDH and GLYAT when comparing tumor vs. non-tumor in LIHC (**C**) and LIRI-JP (**D**). (**E**,**F**) Survival analyses of patients in LIHC-training, LIHC-validation and LIRI-JP-testing cohorts when dividing the population into high and low ssGSEA scores for ABAT (**E**), DMGDH (**F**) and GLYAT signatures.

**Figure 4 biomolecules-14-00653-f004:**
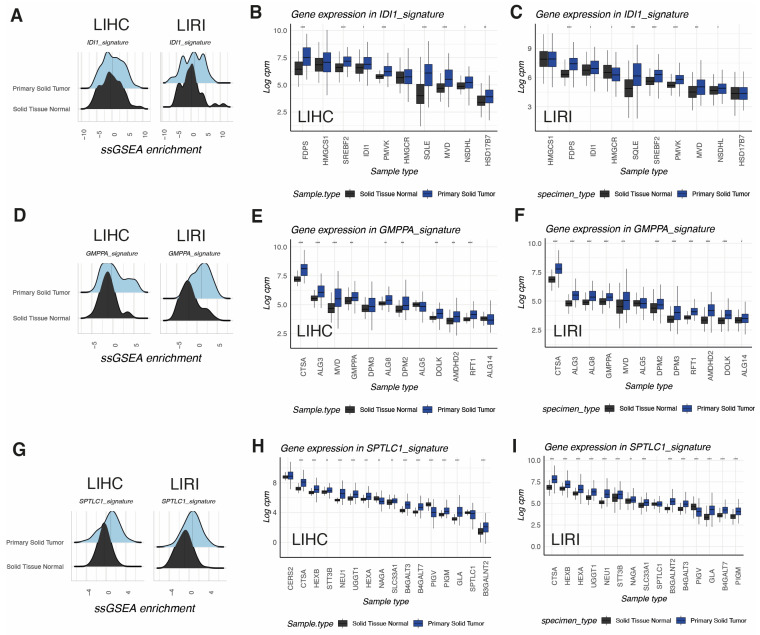
IDI1, GMPPA, and SPTLC1-centered clusters are overexpressed metabolic signatures in HCC. (**A**) Ridge plots of ssGSEA scores of Isopentenyl-diphosphate delta-isomerase 1 (IDI1) signature in LIHC and LIRI-JP cohorts compared with paired non-tumor tissue. (**B**,**C**) Box plots showing the expression levels of individual genes included in the IDI1 signature in LIHC (**B**) and LIRI-JP (**C**) cohorts. (**D**) Ridge plots of ssGSEA scores of GDP-mannose pyrophosphorylase A (GMPPA) signature in LIHC and LIRI-JP cohorts compared with paired non-tumor tissue. (**E**,**F**) Box plots showing the expression levels of individual genes included in GMPPA signature in LIHC (**E**) and LIRI-JP (**F**) cohorts. (**G**) Ridge plots of ssGSEA scores of Serine Palmitoyltransferase Long-Chain Base Subunit 1 (SPTLC1) signature in the LIHC and LIRI-JP cohorts compared with paired non-tumor tissue. (**H**,**I**) Box plots showing the expression levels of individual genes included in GMPPA signature in the LIHC (**H**) and LIRI-JP (**I**) cohorts. Abbreviations: * *p* < 0.05, ** *p* < 0.01, *** *p* < 0.001, **** *p* < 0.0001.

**Figure 5 biomolecules-14-00653-f005:**
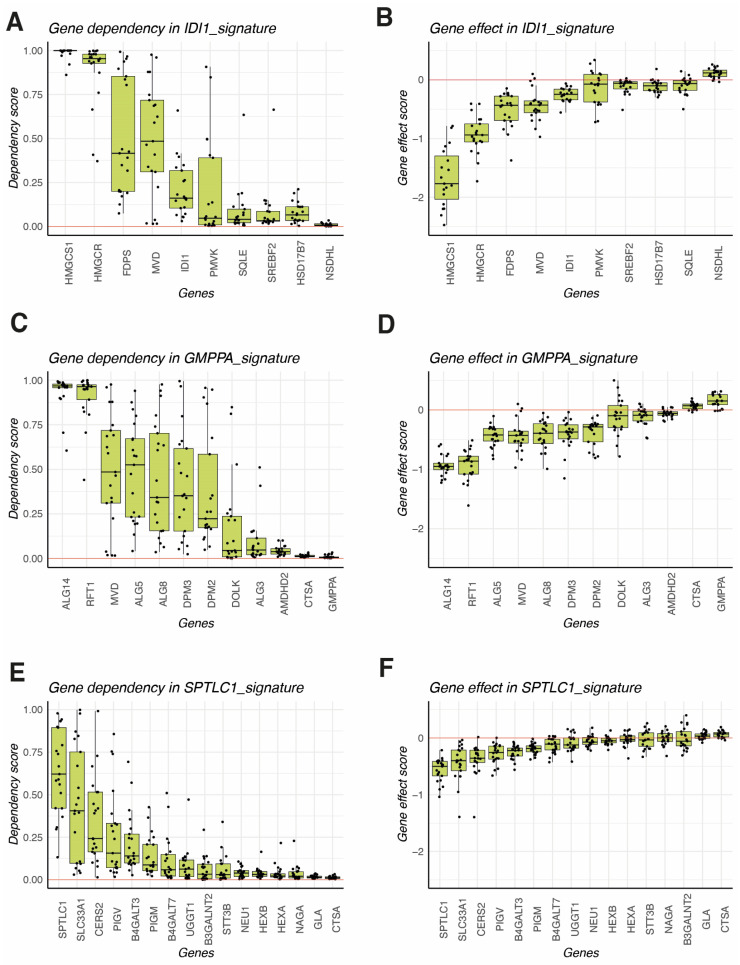
Identification of metabolic vulnerabilities in HCC cell lines using DepMap. (**A**) Gene dependency scores for individual genes included in the IDI1 signature. (**B**) Gene effect scores for individual genes included in the IDI1 signature. (**C**) Gene dependency scores for individual genes included in the GMPPA signature. (**D**) Gene effect scores for individual genes included in the GMPPA signature. (**E**) Gene dependency scores for individual genes included in the SPTLC1 signature. (**F**) Gene effect scores for individual genes included in the SPTLC1 signature.

**Figure 6 biomolecules-14-00653-f006:**
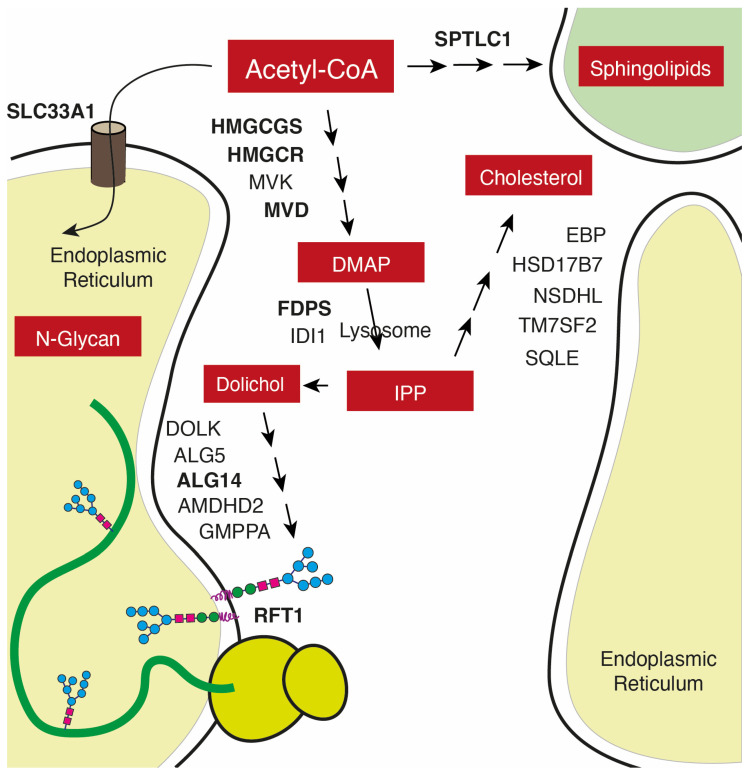
Theoretical model depicting the N-glycan, mevalonate, and sphingolipid biosynthetic pathways as integrated oncometabolic responses lead to a shift in Acetyl CoA use in HCC.

**Table 1 biomolecules-14-00653-t001:** Main metabolic signatures upon hierarchical clustering of the TCGA-LIHC cohort.

Hierarchical Cluster	Signature Name	Function
Group 1 Liver-specific	CYP4A22, CPT2	Fatty acid metabolism and transport
	GCDH	Lysine, hydroxylysine, and tryptophan metabolism
	SARDH	Glycine cleavage
	AASS	Lysine catabolism
	MCCC2	Leucine catabolism
	SDHA, SDHB	Respiratory chain reaction
	CYP3A4, AOX1	Detoxification and metabolism of xenobiotics
Group 2 ECM metabolism	LUM, DCN, VCAN, THBS2, COL3A1	Extracellular matrix, organization, and cell adhesion
	ALOXAP5	Leukotriene metabolism
	CHST3, B3GALT5	Modification of glucosamine glycans
Group 3 Proliferation	NME1	Nucleotide metabolism
	RPL37A	Protein synthesis
	COX5B	Oxidative phosphorylation
	GPX4	Glutathione management
	GMPPA	Cytoplasmic glycosylation
Group 4 Cholesterol	IDI1, FDPS, DHCR7, EBP	Mevalonate and cholesterol biosynthesis
Group 5	EP300	Nuclear factors
	PIK3C2A	Inositol phosphate metabolism
Unclustered	DLD	Glycolysis
	SPTLC1	Sphingolipid metabolism
	ABCC2	Glucuronidation and transport of bilirubin
	NAT2	Metabolism of drugs

## Data Availability

The scripts used to adapt public signatures to an expression matrix using graph-based statistics models are available from GitHub (https://github.com/unav-hcclab/gsadapt/blob/6b7e7ed083882c22b64c01b9318cb57087d45c40/gsadapt_pipeline, accessed on 27 April 2024).

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
