# Peer review of "Application of Graph Models to the Identification of Transcriptomic Oncometabolic Pathways in Human Hepatocellular Carcinoma"

_biomolecules, 2024, doi:10.3390/biom14060653_

Round 1

Reviewer 1 Report

Comments and Suggestions for Authors

Barace and colleagues interrogated publicly available databases of HCC and healthy liver and generated new metabolism-related gene signatures. They correlated the new metabolic signatures with HCC drivers and with previously described HCC subtypes. Importantly, the new generated metabolic gene signatures have prognosis value, they are validated in an independent database of HCC and by using the DepMap database.

The manuscript is well written and the results are important. This new metabolic signatures can be used in the future to design better/personalized treatments for HCC and/or predict response to actual treatments.

Major point

I wonder how specific are these new metabolic gene signatures for HCC. For example, the NME1 gene signature looks to me that could be present in most of the tumors, independently of the tumor origin.

Author Response

Barace and colleagues interrogated publicly available databases of HCC and healthy liver and generated new metabolism-related gene signatures. They correlated the new metabolic signatures with HCC drivers and with previously described HCC subtypes. Importantly, the new generated metabolic gene signatures have prognosis value, they are validated in an independent database of HCC and by using the DepMap database.

The manuscript is well written and the results are important. This new metabolic signatures can be used in the future to design better/personalized treatments for HCC and/or predict response to actual treatments.

Major point

I wonder how specific are these new metabolic gene signatures for HCC. For example, the NME1 gene signature looks to me that could be present in most of the tumors, independently of the tumor origin.

RESPONSE TO REVIEWER #1

We appreciate the comments of the reviewer. We agree with the comment about NME1 signature. We have added a new sentence in the discussion, acknowledging that this signature is not necessarily HCC-specific. Actually, we can not say that the signatures uncovered in the paper are HCC-specific, as suggested by the Reviewer. Nevertheless, a metabolic dissection of HCC compared with non-tumoral liver, has led to identifying new metabolic vulnerabilities. These could be equally important in other cancers and we will explore this possibility in the future.

Reviewer 2 Report

Comments and Suggestions for Authors

In this manuscript, the authors use several bioinformatic approaches to identify the presence of the presence of TP53 or CTNNB1 driver mutations in Liver cohort. In addition, the authors identified genes that were related to the metabolic domains that may not impact therapeutic development. Lastly, they also used validation cohort and method to verify specific metabolic pathways that were important in liver.

Specific comments:

1.The study utilised graph topologies or visualisation of the graph to filter the specific biologically-relevant features for downstream work. The authors should remove “graph models” to avoid confusion. Graph models often refer to neural model, machine learning model, AI model to construct graphs, in which these models were not used in their work.

2. The results were confusing. It was unclear whether all samples (i.e. patients, cell cultures) downloaded from public database had liver hepatocellular carcinoma?

3. Were any of these patients with hepatocellular carcinoma had Hepatitis C Virus infection? As stated in the literatures, hepatitis C virus infection is a major cause of hepatocellular carcinoma.

4. In the Methods section, the authors did not include the parameters that were used for the bioinformatic analyses. For instance, in GSEA, what were Max Size and Min Size? Several bioinformatics approaches were described but parameters were not included.

5. Liver generally has an immunosuppressive environment as compared to other organs. Can the authors further discussed if immunotherapy will work on these patients?

Comments on the Quality of English Language

Please correct grammar.

Author Response

Comments from Reviewer #2

Point-by-point reply.

  1. The study utilised graph topologies or visualisation of the graph to filter the specific biologically-relevant features for downstream work. The authors should remove “graph models” to avoid confusion. Graph models often refer to neural model, machine learning model, AI model to construct graphs, in which these models were not used in their work.
    • We thank the reviewer for pointing out the potential misunderstanding with the terminology. We agree with the reviewer and the text now reads graph statistics rather than graph models
  2. The results were confusing. It was unclear whether all samples (i.e. patients, cell cultures) downloaded from public database had liver hepatocellular carcinoma?
    • Indeed we have only studied HCC from the TCGA-LIHC and the LIRI-JP cohort. We have stressed this fact that probably was not completely clear in the methods and results section.
  3. Were any of these patients with hepatocellular carcinoma had Hepatitis C Virus infection? As stated in the literatures, hepatitis C virus infection is a major cause of hepatocellular carcinoma.
    • We agree with the reviewer, that HCV was the most important cause of HCC until the upcoming of the new direct antiviral drugs a decade ago. Our cohorts are complementary in this regard: a total of 18 patients of the TCGA-LIHC (4.93%), and 113 patients of the LIRI (56.50%), had chronic HCV infection. This result is mentioned in the supplementary table 1, together with the remaining clinical characteristics of both cohorts. We have put additional mention to this in the methods section.
  4. In the Methods section, the authors did not include the parameters that were used for the bioinformatic analyses. For instance, in GSEA, what were Max Size and Min Size? Several bioinformatics approaches were described but parameters were not included.
    • We thank the reviewer for this observation. We have added the parameters specifying ssGSEA and other methods. The whole Methods section has been reviewed to improve the readability and precision.
  5. Liver generally has an immunosuppressive environment as compared to other organs. Can the authors further discussed if immunotherapy will work on these patients?
    • We completely agree with the reviewer. We actually know immunotherapy works in around 50% of patients (Control Disease Rate) and generates objective responses in around 30% (Objective Response Rate), including Complete responses in around 8-10%. The currently approved schemes are Atezolizumab+Bevacizumab and Tremelimumab+Durvalumab under STRIDE protocol. Unfortunately the cohorts where these drugs were tested in phase III clinical trials (IMbrave150 and Himalaya Trials), have not yet released tumor RNAseq studies where we can search for metabolic signatures. We have added a comment to the discussion section.

Comments on the Quality of English Language

Please correct grammar.

  • We appreciate the note from the reviewer, we have done a deep review of our English grammar.